# BLIP3-O: A FAMILY OF FULLY OPEN UNIFIED MULTIMODAL MODELS—ARCHITECTURE, TRAINING AND DATASET

## ABSTRACT

Unifying image understanding and generation has gained growing attention in recent research on multimodal models. Although design choices for image understanding have been extensively studied, the optimal model architecture and training recipe for a unified framework with image generation remain underexplored. Motivated by the strong potential of autoregressive and diffusion models for high-quality generation and scalability, we conduct a comprehensive study of their use in unified multimodal settings, with emphasis on image representations, modeling objectives, and training strategies. Grounded in these investigations, we introduce a novel approach that employs a diffusion transformer to generate semantically rich CLIP image features, in contrast to conventional VAE-based representations. This design yields both higher training efficiency and improved generative quality. Furthermore, we demonstrate that a sequential pretraining strategy for unified models—first training on image understanding and subsequently on image generation—offers practical advantages by preserving image-understanding capability while developing strong image generation ability. Finally, we carefully curate a high-quality instruction-tuning dataset BLIP3o-60k for image generation by prompting GPT-4o with a diverse set of captions covering various scenes, objects, human gestures, and more. Building on our innovative model design, training recipe, and datasets, we develop BLIP3-O, a suite of state-of-the-art unified multimodal models. BLIP3-O achieves superior performance across most of the popular benchmarks spanning both image understanding and generation tasks. *To facilitate future research, we fully open-source our models, including code, model weights, training scripts, and pretraining and instruction tuning datasets.*

## 1 INTRODUCTION

Recent advances have demonstrated the potential for unified multimodal representation learning that supports both image understanding and image generation within a single model (Ge et al., 2024; Sun et al., 2024; Xie et al., 2024; Wu et al., 2024a; Chen et al., 2025; Tong et al., 2024; Pan et al., 2025). In this field, despite extensive studies on image understanding, the optimal architecture and training strategy for image generation remain underexplored. The previous debate revolves around two approaches: the first approach quantizes continuous visual features into discrete tokens and models them as a categorical distribution (Team, 2024; Wang et al., 2024; Ma et al., 2025); the second approach generates intermediate visual features or latent representations via the autoregressive model and then conditions on these visual features to generate images through the diffusion model (Tong et al., 2024; Pan et al., 2025). The recently released GPT-4o image generation (gpt, 2025) was implied to adopt a hybrid architecture with autoregressive and diffusion models following the second approach (gpt, 2025; Yan et al., 2025). The wide range of image representations used in industry inspired our systematic study of design choices in unified models. Specifically, our investigation focuses on three key design axes: (1) **image representations** – whether to encode the images into low-level pixel features (e.g., from VAE-based encoders) or high-level semantic features (e.g., from CLIP image encoders); (2) **training objectives** – Mean Squared Error (MSE) versus Flow Matching (Lipman et al., 2022b; Liu et al., 2022), and what their impacts on training efficiency and generation quality; (3) **training strategies** – one can use simultaneous multitask training on image

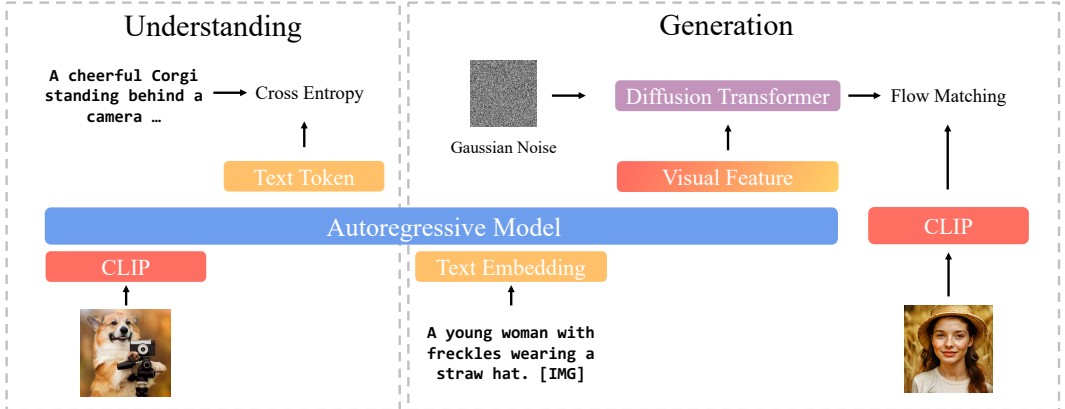

Figure 1: The architecture of BLIP3-O. For image understanding part, we use CLIP to encode the image and compute the cross entropy loss between the target text token and predicted text token. For image generation part, autoregressive model first generates a sequence of intermediate visual features, which are then used as conditioning inputs to a diffusion transformer that generates CLIP image features to approximate the ground-truth CLIP features. By using CLIP encoder, image understanding and image generation share the same semantic space, effectively unifying these two tasks.

understanding and generation like Metamorph (Tong et al., 2024), or else use sequential training like LMFusion (Shi et al., 2024) and MetaQuery (Pan et al., 2025), in which the model is first trained for understanding and then extended for generation.

Our findings reveal that CLIP image features offer more compact and informative representations than VAE features, resulting in both faster training and higher image generation quality. Flow matching loss proves to be more effective than MSE loss, enabling more diverse image sampling and yielding better image quality. Furthermore, we find that a sequential training strategy—first training the autoregressive model on image understanding tasks, then freezing it during training on image generation—achieves the best overall performance. Based on these findings, we develop BLIP3-O, a herd of state-of-the-art unified multimodal models. BLIP3-O leverages diffusion transformer and flow matching on CLIP features (Figure 1) and is sequentially trained on image understanding and image generation tasks. To further improve visual aesthetic and instruction following abilities, we carefully curate a 60k high-quality instruction-tuning dataset BLIP3o-60k for image generation, by prompting GPT-4o with a diverse set of prompts spanning scenes, objects, human gestures and more. We observe that supervised instruction tuning on BLIP3o-60k significantly enhances the alignment of BLIP3-O with human preference and improves the aesthetic quality.

In our experiments, BLIP3-O achieves superior performance across most of the popular benchmarks for image understanding and image generation. *To support further research and keep the mission of open-source foundation model research like BLIP-3 (Xue et al., 2024), we fully open-source our models including model weights, code, pretraining and instruction-tuning datasets, and evaluation pipelines. We hope that our work will support the research community and drive continued progress in the unified multimodal domain.*

## 2 UNIFIED MULTIMODAL FOR IMAGE GENERATION AND UNDERSTANDING

Recent OpenAI's GPT-4o (gpt, 2025) has demonstrated state-of-the-art performance in image understanding, generation and editing tasks. Emerging hypotheses of its architecture (Yan et al., 2025) suggest a hybrid pipeline structured as:

$$\textbf{Tokens} \longrightarrow \textbf{[Autoregressive Model]} \longrightarrow \textbf{[Diffusion Model]} \longrightarrow \textbf{Image Pixels}$$

indicating that autoregressive and diffusion models may be jointly leveraged to combine the strengths of both modules. Motivated by this hybrid design, we adopt an autoregressive + diffusion framework in our study. However, the optimal architecture in this framework remains unclear. The autoregressive model produces continuous intermediate visual features intended to approximate ground-truth image representations, raising two key questions. First, what should serve as the ground-truth embeddings:

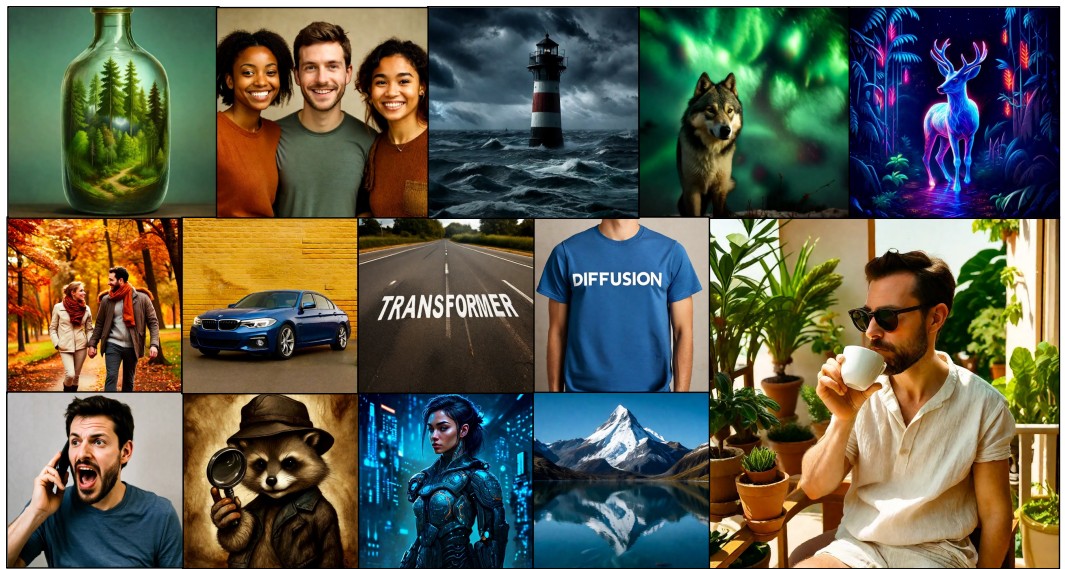

Figure 2: Visualization results of BLIP3-O 8B at 1024×1024 resolution.

should we use a VAE or CLIP to encode images into continuous features? Second, once the autoregressive model generates visual features, how do we optimally align them with the ground-truth image features, or more generally, how should we model the distribution of these continuous visual features: via a simple MSE loss, or by employing a diffusion-based approach? Thus, we conduct a comprehensive exploration of various design choices in the following section.

## 3 IMAGE GENERATION IN UNIFIED MULTIMODAL

In this section, we discuss the design choices involved in building the image generation model within a unified multimodal framework. We begin by exploring how images can be represented as continuous embeddings through encoder–decoder architectures, which play a foundational role in learning efficiency and generation quality.

### 3.1 IMAGE ENCODING AND RECONSTRUCTION

Image generation typically begins by encoding an image into a continuous latent embedding using an encoder, followed by a decoder that reconstructs the image from this latent embedding. This encoding-decoding pipeline can effectively reduce the dimensionality of the input space in image generation, facilitating efficient training. In the following, we discuss two widely used encoder–decoder paradigms.

**Variational Autoencoders**  Variational Autoencoders (VAEs) (Kingma et al., 2013; Rezende et al., 2014) are a class of generative models that learn to encode images into a structured, continuous latent space. The encoder approximates the posterior distribution over the latent variables given the input image, while the decoder reconstructs the image from samples drawn from this latent distribution. Latent diffusion models build on this framework by learning to model the distribution of compressed latent representations, rather than raw image pixels. By operating in the VAE latent space, these models significantly reduce the dimensionality of the output space, thereby lowering computational costs and enabling more efficient training. After the denoising steps, the VAE decoder maps the generated latent embeddings into raw image pixels.

**CLIP Encoder with Diffusion Decoder**  CLIP (Radford et al., 2021) models have become foundational encoders for image understanding tasks (Liu et al., 2023a), owing to its strong ability to extract rich, high-level semantic features from images through contrastive training on large-scale image–text pairs. However, leveraging these features for image generation remains a non-trivial challenge,

as CLIP was not originally designed for reconstruction tasks. Emu2 (Sun et al., 2024) presents a practical solution by pairing a CLIP-based encoder with a diffusion-based decoder. Specifically, it uses EVA-CLIP to encode images into continuous visual embeddings and reconstructs them via a diffusion model initialized from SDXL-base (Podell et al., 2023). During training, the diffusion decoder is fine-tuned to use the visual embeddings from EVA-CLIP as conditions to recover the original image from Gaussian noise, while the EVA-CLIP remains frozen. This process effectively combines the CLIP and diffusion models into an image autoencoder: the CLIP encoder compresses an image into semantically rich latent embeddings, and the diffusion-based decoder reconstructs the image from these embeddings. Notably, although the decoder is based on diffusion architecture, it is trained with a reconstruction loss rather than probabilistic sampling objectives. Consequently, during inference, the model performs deterministic reconstruction.

**Discussion**  These two encoder–decoder architectures, i.e., VAEs and CLIP-Diffusion, represent distinct paradigms for image encoding and reconstruction, each offering specific advantages and trade-offs. VAEs encode the image into low-level pixel features and offer better reconstruction quality. Furthermore, VAEs are widely available as off-the-shelf models and can be integrated directly into image generation training pipelines. In contrast, CLIP-Diffusion requires additional training to adapt the diffusion models to various CLIP encoders. However, CLIP-Diffusion architectures offer significant benefits in terms of image compression ratio. For example, in both Emu2 (Sun et al., 2024) and our experiments, each image regardless of its resolution can be encoded into a fixed length of 64 continuous vectors, providing both compact and semantically rich latent embeddings. By contrast, VAE-based encoders tend to produce a longer sequence of latent embeddings for higher-resolution inputs, which increases the computational burden in the training procedure.

## 3.2 Modeling Latent Image Representation

After obtaining continuous image embeddings, we proceed to model them using autoregressive architectures. Given a user prompt (e.g., *"A young woman with freckles wearing a straw hat."*), we first encode the prompt into a sequence of embedding vectors $\mathbf{C}$ using the autoregressive model's input embedding layer, and append a learnable query vector $\mathbf{Q}$ to $\mathbf{C}$, where $\mathbf{Q}$ is randomly initialized and optimized during training. As the combined sequence $[\mathbf{C}; \mathbf{Q}]$ is processed through the autoregressive transformer, $\mathbf{Q}$ learns to attend to and extract relevant semantic information from the prompt $\mathbf{C}$. The resulting $\mathbf{Q}$ is interpreted as the intermediate visual features or latent representation generated by the autoregressive model, and is trained to approximate the ground-truth image feature $\mathbf{X}$ (obtained from VAE or CLIP). In the following, we introduce two training objectives: Mean Squared Error (MSE) and Flow Matching, for learning to align $\mathbf{Q}$ with the ground-truth image embedding $\mathbf{X}$.

**MSE Loss**  The Mean Squared Error (MSE) loss is a straightforward and widely used objective for learning continuous image embeddings (Ge et al., 2024; Sun et al., 2024). Given the predicted visual features $\mathbf{Q}$ produced by the autoregressive model and the ground-truth image features $\mathbf{X}$, we first apply a learnable linear projection to align the dimensionality of $\mathbf{Q}$ with that of $\mathbf{X}$. The MSE loss is then formulated as:

$$\mathcal{L}_{\text{MSE}} = \|\mathbf{X} - \mathbf{W}\mathbf{Q}\|_2^2,$$

where $\mathbf{W}$ denotes the learnable projection matrix.

**Flow Matching**  Note that using MSE loss only aligns the predicted image features $\mathbf{Q}$ with the mean value of the target distribution. An ideal training objective would model the probability distribution of continuous image representation. We propose to use flow matching (Lipman et al., 2022a), a diffusion framework that can sample from a target continuous distribution by iterative transporting samples from a prior distribution (e.g., Gaussian). Given a ground-truth image feature $\mathbf{X}_1$ and the condition $\mathbf{Q}$ encoded by an autoregressive model, at each training step, we sample a timestep $t \sim \mathcal{U}(0, 1)$, and noise $\mathbf{X}_0 \sim \mathcal{N}(0, 1)$. Then diffusion transformer learns to predict the velocity $\mathbf{V}_t = \frac{d\mathbf{X}_t}{dt}$ at the timestep $t$ conditioned on $\mathbf{Q}$, in the direction of $\mathbf{X}_1$. Following previous work (Liu et al., 2022), we compute $\mathbf{X}_t$ by a simple linear interpolation between $\mathbf{X}_0$ and $\mathbf{X}_1$:

$$\mathbf{X}_t = t\mathbf{X}_t + (1 - t)\mathbf{X}_0,$$

and the analytical solution of $\mathbf{V}_t$ can be expressed as:

$$\mathbf{V}_t = \frac{d\mathbf{X}_t}{dt} = \mathbf{X}_t - \mathbf{X}_0.$$

Finally, the training objective is defined as:

$$\mathcal{L}_{\text{Flow}}(\theta) = \mathbb{E}_{(\mathbf{X}_1,\mathbf{Q})\sim\mathcal{D},t\sim\mathcal{U}(0,1),\mathbf{X}_0\sim\mathcal{N}(0,1)} \left[\|\mathbf{V}_\theta(\mathbf{X}_t,\mathbf{Q},t) - \mathbf{V}_t\|^2\right],$$

where $\theta$ is the diffusion transformer's parameters, and $\mathbf{V}_\theta(\mathbf{X}_t,\mathbf{Q},t)$ denotes the predicted velocity based on an instance $(\mathbf{X}_1,\mathbf{Q})$, timestep $t$, and noise $\mathbf{X}_0$.

**Discussion** Unlike discrete tokens, which inherently support sampling-based strategies for exploring diverse generation paths, continuous representations lack this property. Specifically, under an MSE-based training objective, the predicted visual features $\mathbf{Q}$ become nearly deterministic for a given prompt. As a result, the output images, regardless of whether the visual decoder is based on VAEs or CLIP + Diffusion architectures, remain almost identical across multiple inference runs. This determinism highlights a key limitation of the MSE objective: it constrains the model to produce a single, fixed output for each prompt, thereby limiting generation diversity. In contrast, the flow matching framework enables the model to inherit the stochasticity of the diffusion process. This allows the model to generate diverse image samples conditioned on the same prompt, facilitating broader exploration of the output space. However, this flexibility comes at the cost of increased model complexity. Flow matching introduces additional learnable parameters compared to MSE. In our implementation, we use a diffusion transformer (DiT), and empirically find that scaling its capacity yields significant performance improvements.

### 3.3 DESIGN CHOICES

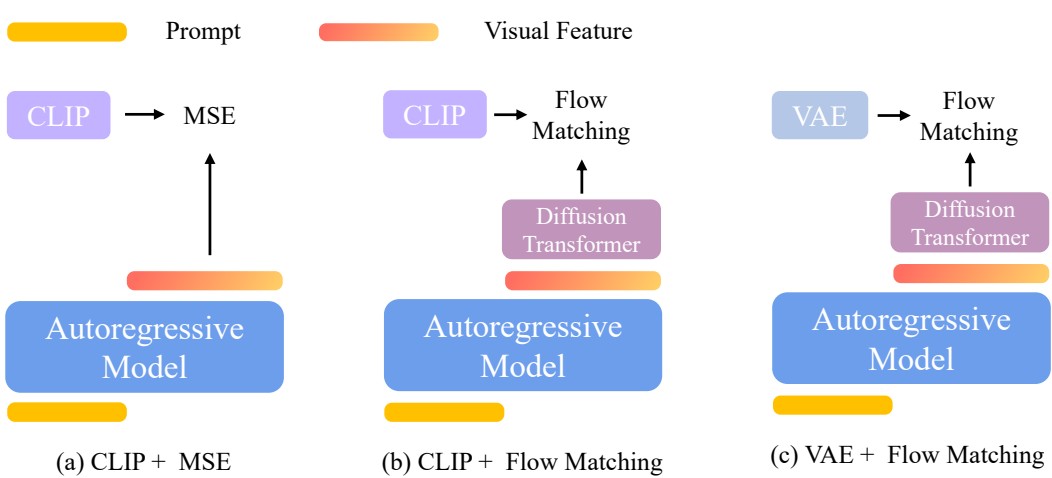

(a) CLIP + MSE  (b) CLIP + Flow Matching  (c) VAE + Flow Matching

Figure 3: Three design choices for image generation in unified multimodal model. All designs use a **Autoregressive + Diffusion** framework but vary in their image generation components. For the flow matching loss, we keep the autoregressive model frozen and only fine-tune the image generation module to preserve the model's language capabilities.

The combination of different image encoder–decoder architectures and training objectives gives rise to a range of design choices for image generation models. These design choices, illustrated in Figure 3, significantly influence both the quality and controllability of the generated images. In this section, we summarize and analyze the trade-offs introduced by different encoder types (e.g., VAEs vs. CLIP encoders) and loss functions (e.g., MSE vs. Flow Matching).

**CLIP + MSE** Following Emu2 (Sun et al., 2024), Seed-X (Ge et al., 2024) and Metamorph (Tong et al., 2024), we use CLIP to encode images into 64 fixed-length semantic-rich visual embeddings. The autoregressive model is trained to minimize the Mean Squared Error (MSE) loss between the predicted visual features $\mathbf{Q}$ and the ground-truth CLIP embedding $\mathbf{X}$, as illustrated in Figure 3(a). During inference, given a text prompt $\mathbf{C}$, the autoregressive model predicts the latent visual features $\mathbf{Q}$, which is subsequently passed to a diffusion-based visual decoder to reconstruct the real image.

**CLIP + Flow Matching** As an alternative to MSE loss, we employ flow matching loss to train the model to predict ground-truth CLIP embeddings, as illustrated in Figure 3(b). Given a prompt **C**, the autoregressive model generates a sequence of visual features **Q**. These features are used as conditions to guide the diffusion process, yielding a predicted CLIP embedding to approximate the ground-truth CLIP features. In essence, the inference pipeline involves two diffusion stages: the first uses the conditioning visual features **Q** to iteratively denoise into CLIP embeddings. And the second converts these CLIP embeddings into real images by diffusion-based visual decoder. This approach enables stochastic sampling at the first stage, allowing for greater diversity in image generation.

**VAE + Flow Matching** We can also use flow matching loss to predict the ground truth VAE features seen in Figure 3(c), which is similar to MetaQuery (Pan et al., 2025). At inference time, given a prompt **C**, the autoregressive model produces visual features **Q**. Then, conditioning on **Q** and iteratively removing noise at each step, the real images are generated by the VAE decoder.

**VAE + MSE** Because our focus is on autoregressive + diffusion framework, we exclude VAE + MSE approaches, as they do not incorporate any diffusion module.

**Implementation Details** To compare various design choices, we use Llama-3.2-1B-Instruct as autoregressive model. Our training data consists of CC12M (Changpinyo et al., 2021), SA-1B (Kirillov et al., 2023), and JourneyDB (Sun et al., 2023), amounting to approximately 25 million samples. The detailed description of image generation architecture using flow matching loss is provided in Section 5.1.

**Results** We report the FID score (Heusel et al., 2017) on MJHQ-30k (Li et al., 2024) for visual aesthetic quality, along with GenEval (Ghosh et al., 2023) and DPG-Bench (Hu et al., 2024) metrics for evaluating prompt alignment. We plot the results for each design choice at approximately every 3,200 training steps. Figure 4 shows that CLIP + Flow Matching achieves the best prompt alignment scores on both GenEval and DPG-Bench, while VAE + Flow Matching produces the lowest (best) FID, indicating superior aesthetic quality. However, FID has inherent limitations: it quantifies stylistic deviation from the target image distribution and often overlooks true generative quality and prompt alignment. In fact, our FID evaluation of GPT-4o on the MJHQ-30k dataset produced a score of around 30.0, underscoring that FID can be misleading in the image generation evaluation. In general, our experiments demonstrate CLIP + Flow Matching as the most effective design choice.

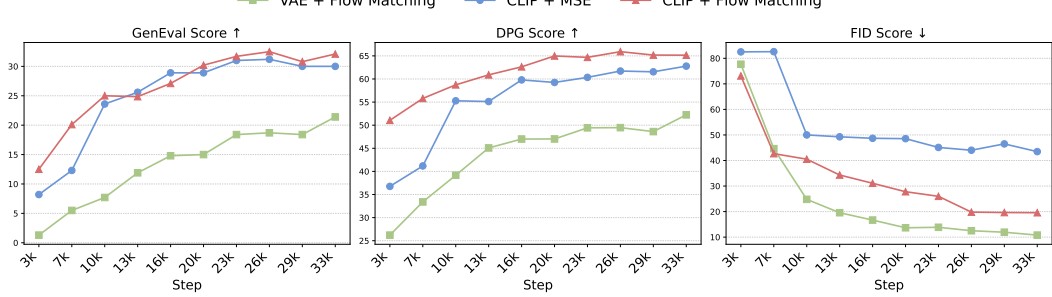

Figure 4: Comparison of different design choices.

**Discussion** In this section, we present a comprehensive evaluation of various design choices for image generation within a unified multimodal framework. Our results clearly show that CLIP's features produce more compact and semantically rich representations than VAE features, yielding higher training efficiency. Autoregressive models more effectively learn these semantic-level features compared to pixel-level features. Furthermore, flow matching proves to be a more effective training objective for modeling the image distribution, resulting in greater sample diversity and enhanced visual quality.

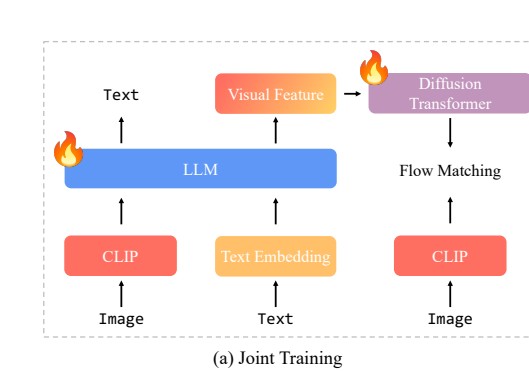 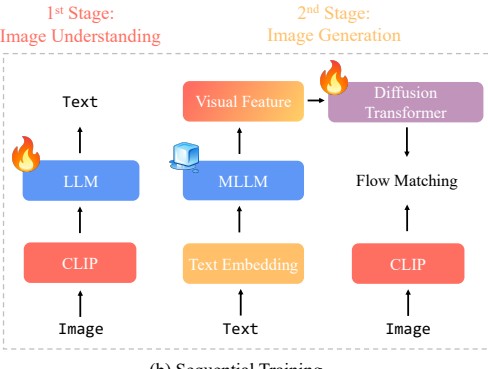

(a) Joint Training        (b) Sequential Training

Figure 5: Joint Training vs. Sequential Training: Joint training performs multitask learning by mixing image-understanding and image-generation data, updating both the autoregressive backbone and the generation module simultaneously. Sequential training separates the process: first, the model is trained only on image-understanding tasks; then the autoregressive backbone is frozen and only the image-generation module is trained in a second stage.

## 4   TRAINING STRATEGIES FOR UNIFIED MULTIMODAL

Building on our image generation study, the next step is to develop a unified model that can perform both image understanding and image generation. We use CLIP + Flow Matching for the image generation module. Since image understanding also operates in CLIP's embedding space, we align both tasks within the same semantic space, enabling their unification. In this context, we discuss two training strategies, as shown in Figure 5, to achieve this integration.

### 4.1   JOINT TRAINING VERSUS SEQUENTIAL TRAINING

**Joint Training**   Joint training of image understanding and image generation has become a common practice in recent works such as Metamorph (Tong et al., 2024), Janus-Pro (Chen et al., 2025), and Show-o (Xie et al., 2024). Although these methods adopt different architectures for image generation, all perform multitask learning by mixing data for image generation and understanding.

**Sequential Training**   Instead of training image understanding and generation together, we follow a two-stage approach. In the first stage, we train only the image understanding module. In the second stage, we freeze the MLLM backbone, and train only the image generation module.

Table 1: Results on different training strategies.

| Training Strategy | GenEval | DPG | WISE | VQAv2 | MMBench | SEED | MM-Vet |
|---|---|---|---|---|---|---|---|
| Joint | 0.74 | 73.59 | 46.18 | 70.0 | 74.1 | 70.7 | 52.8 |
| Sequential | 0.77 | 75.24 | 47.72 | 75.9 | 78.6 | 73.8 | 60.1 |

### 4.2   DISCUSSION

In a joint training setting, although image understanding and data generation tasks possibly benefit each other as demonstrated by Metamorph (Tong et al., 2024), two critical factors influence their synergistic effect: (i) the total size of training data and (ii) the data ratio between image understanding and data generation. In Table 1, we adopt a 1:1 ratio between understanding and generation data. In this setting, we observe that sequential training consistently outperforms joint training in all evaluation metrics. Therefore, we adopt sequential training to build our unified multimodal model.

## 5 BLIP3-O: OUR STATE-OF-THE-ART UNIFIED MULTIMODAL

### 5.1 MODEL ARCHITECTURE

Based on our findings, we adopt **CLIP + Flow Matching** and **sequential training** to train unified multimodal model BLIP3-O. We develop two different size models: 4B and 8B based on Qwen 2.5 VL (Bai et al., 2025). In the 8B model, we freeze the Qwen2.5-VL-7B-Instruct backbone and train the diffusion transformers, totaling 1.4 B trainable parameters. The 4B model follows the same image generation architecture but uses Qwen2.5-VL-3B-Instruct as backbone. More details about the model architecture can be seen in the Appendix.

### 5.2 TRAINING RECIPE

**Stage 1: Pretraining for Image Generation**  We use 55 million image in the pretraining stage, each image is paired with a detailed caption. To improve generalization to varying prompt lengths, we also include around 10% (6 million) shorter captions.

**Stage 2: Instruction Tuning for Image Generation**  After pretraining stage, we observe several failures in the model generation: complex human gestures, knowledge-based objects and simple text. To remedy this, we perform instruction tuning focused specifically on these domains. For each category, we prompt GPT-4o to generate roughly 10k prompt–image pairs, creating a targeted dataset that improves the model's ability to handle these cases. To improve visual aesthetics quality, we also expand our data with prompts drawn from JourneyDB (Sun et al., 2023) and DALL·E 3. This process yields a curated collection of approximately 60k high quality prompt–image pairs.

### 5.3 RESULTS

For baseline comparison, we include the following unified models: EMU2 Chat (Sun et al., 2024), Chameleon (Team, 2024), Seed-X (Ge et al., 2024), VILA-U (Wu et al., 2024b), LMfusion (Shi et al., 2024), Show-o (Xie et al., 2024), EMU3 (Wang et al., 2024), MetaMorph (Tong et al., 2024), TokenFlow (Qu et al., 2024), Janus (Wu et al., 2024a), and Janus-Pro (Chen et al., 2025).

**Image Understanding**  In the image understanding task, we evaluate the benchmark performance on VQAv2 (Goyal et al., 2017), MMBench (Liu et al., 2023b), SeedBench (Li et al., 2023), MM-Vet (Yu et al., 2023), MME-Perception and MME-Cognition (Fu et al., 2024), MMMU (Yue et al., 2024), TextVQA (Singh et al., 2019), and RealWorldQA (x.ai, 2023). As shown in Table 2, our BLIP3-O 8B achieves the best performance in most benchmarks.

Table 2: Results on image understanding benchmarks. We highlight the best results in **bold**.

| Model | VQAv2 | MMBench | SEED | MM-Vet | MME-P | MME-C | MMMU | RWQA | TEXTVQA |
|---|---|---|---|---|---|---|---|---|---|
| EMU2 Chat 34B | - | - | 62.8 | 48.5 | - | - | 34.1 | - | 66.6 |
| Chameleon 7B | - | 19.8 | 27.2 | 8.3 | 202.7 | - | 22.4 | 39.0 | 0.0 |
| Chameleon 34B | - | 32.7 | - | 9.7 | 604.5 | - | 38.8 | 39.2 | 0.0 |
| Seed-X 17B | 63.4 | 70.1 | 66.5 | 43.0 | 1457.0 | - | 35.6 | - | - |
| VILA-U 7B | 79.4 | 66.6 | 57.1 | 33.5 | 1401.8 | - | 32.2 | 46.6 | 48.3 |
| LMFusion 16B | - | - | 72.1 | - | 1603.7 | 367.8 | 41.7 | 60.0 | - |
| Show-o 1.3B | 69.4 | - | - | - | 1097.2 | - | 27.4 | - | - |
| EMU3 8B | 75.1 | 58.5 | 68.2 | 37.2 | 1243.8 | 266.1 | 31.6 | 57.4 | 64.7 |
| MetaMorph 8B | - | 75.2 | 71.8 | - | - | - | 41.8 | 58.3 | 60.5 |
| TokenFlow-XL 14B | 77.6 | 76.8 | 72.6 | 48.2 | 1551.1 | 371.1 | 43.2 | 56.6 | 77.6 |
| Janus 1.3B | 77.3 | 75.5 | 68.3 | 34.3 | 1338.0 | - | 30.5 | - | - |
| Janus Pro 7B | - | 79.2 | 72.1 | 50.0 | 1567.1 | - | 41.0 | - | - |
| BLIP3-O 4B | 75.9 | 78.6 | 73.8 | 63.2 | 1574.3 | 632.9 | 53.1 | 60.4 | 78.0 |
| BLIP3-O 8B | **83.0** | **83.5** | **76.9** | **67.1** | **1685.2** | **647.1** | **58.6** | **69.0** | **84.9** |

**Image Generation**  In the image generation benchmark, we report GenEval (Ghosh et al., 2023) and DPG-Bench (Hu et al., 2024) to measure prompt alignment, WISE (Niu et al., 2025) to evaluate world knowledge reasoning capability. As shown in Table 3, BLIP3-O 8B achieves a GenEval score of 0.84, a WISE score of 0.62, but scores lower on DPG-Bench. Because model-based evaluation for

DPG-Bench can be unreliable, we complement these results with a human study on all DPG-Bench prompts in the next section. Furthermore, we also find our instruction tuning dataset BLIP3o-60k yields immediate gains: using only 60k prompt–image pairs, both prompt alignment and visual aesthetics improve markedly, and many generation artifacts are quickly reduced. Although this instruction tuning dataset cannot fully resolve some difficult cases, such as complex human gestures generation, it nonetheless delivers a substantial boost in overall image quality.

Table 3: Image generation benchmark results.

| Model | GenEval | DPG-Bench | WISE |
|---|---|---|---|
| Chameleon 7B | 0.39 | – | - |
| Seed-X 17B | 0.51 | – | - |
| LLaVAFusion 16B | 0.63 | – | - |
| Show-o 1.3B | 0.68 | 67.27 | 0.35 |
| EMU3 8B | 0.66 | 80.60 | 0.39 |
| TokenFlow-XL 14B | 0.63 | 73.38 | - |
| Janus 1.3B | 0.61 | 79.68 | 0.18 |
| Janus Pro 7B | 0.80 | **84.19** | 0.35 |
| BLIP3-O 4B | 0.81 | 79.36 | 0.50 |
| BLIP3-O 8B | **0.84** | 81.60 | **0.62** |

Figure 6: Human study results for DPG-Bench between Janus Pro and our model.

**Inference Efficiency**   To provide a more straightforward comparison, we measured the end-to-end inference time for BLIP3-O and two representative unified models: EMU3 (Wang et al., 2024) and Janus Pro (Chen et al., 2025) on the same A100 GPU. As shown in the Table 4, BLIP3o achieves better inference efficiency compared to Janus-Pro while generating images at a significantly higher resolution.

**Human Study**   In this section, we conduct a human evaluation comparing BLIP3-O 8B and Janus Pro 7B on about 1,000 prompts drawn from the DPG-Bench. We assess two metrics: visual quality and prompt alignment. The details about each metric can be seen in the Appendix.

Table 4: Comparison of inference time.

| Model | Time per Image ↓ | Output Resolution ↑ |
|---|---|---|
| Janus Pro 7B | 12.5s | 384x384 |
| EMU3 8B | 598.1s | 1024x1024 |
| BLIP3-O 8B | 5.58s | 1024x1024 |

Each metric was assessed in two separate rounds, resulting in roughly 3,000 judgments per criterion. As illustrated in Figure 6, BLIP3-O outperforms Janus Pro on both visual quality and prompt alignment, even though Janus Pro achieves a higher DPG score in Table 3. The $p$-values for Visual Quality and Prompt Alignment are 5.05e-06 and 1.16e-05, respectively, indicating that our model significantly outperforms Janus Pro with high statistical confidence.

# 6    RELATED WORK

Recent studies have highlighted unified multimodal, capable of both image understanding and generation, as a promising avenue of research. For example, SEED-X (Ge et al., 2024), Emu-2 (Sun et al., 2024), and MetaMorph (Tong et al., 2024) train image features via regression losses, while Chameleon (Team, 2024), Show-o (Xie et al., 2024), EMU3 (Wang et al., 2024), and Janus (Wu et al., 2024a; Chen et al., 2025) adopt an autoregressive discrete token prediction paradigm. Concurrent work MetaQuery (Pan et al., 2025) also uses learnable queries to bridge frozen pre-trained MLLMs with pre-trained diffusion models, but the diffusion models are in VAE + Flow Matching strategy instead of the more efficient CLIP + Flow Matching one in our BLIP3-O.

# 7    CONCLUSION

We present the first systematic exploration of hybrid autoregressive and diffusion architectures for unified multimodal modeling, evaluating three critical aspects: image representation, training objective and training strategy. Building on these insights, we introduce BLIP3-O, a family of state-of-the-art unified models enhanced with a 60k instruction tuning dataset BLIP3o-60k that substantially improves prompt alignment and visual aesthetics.

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

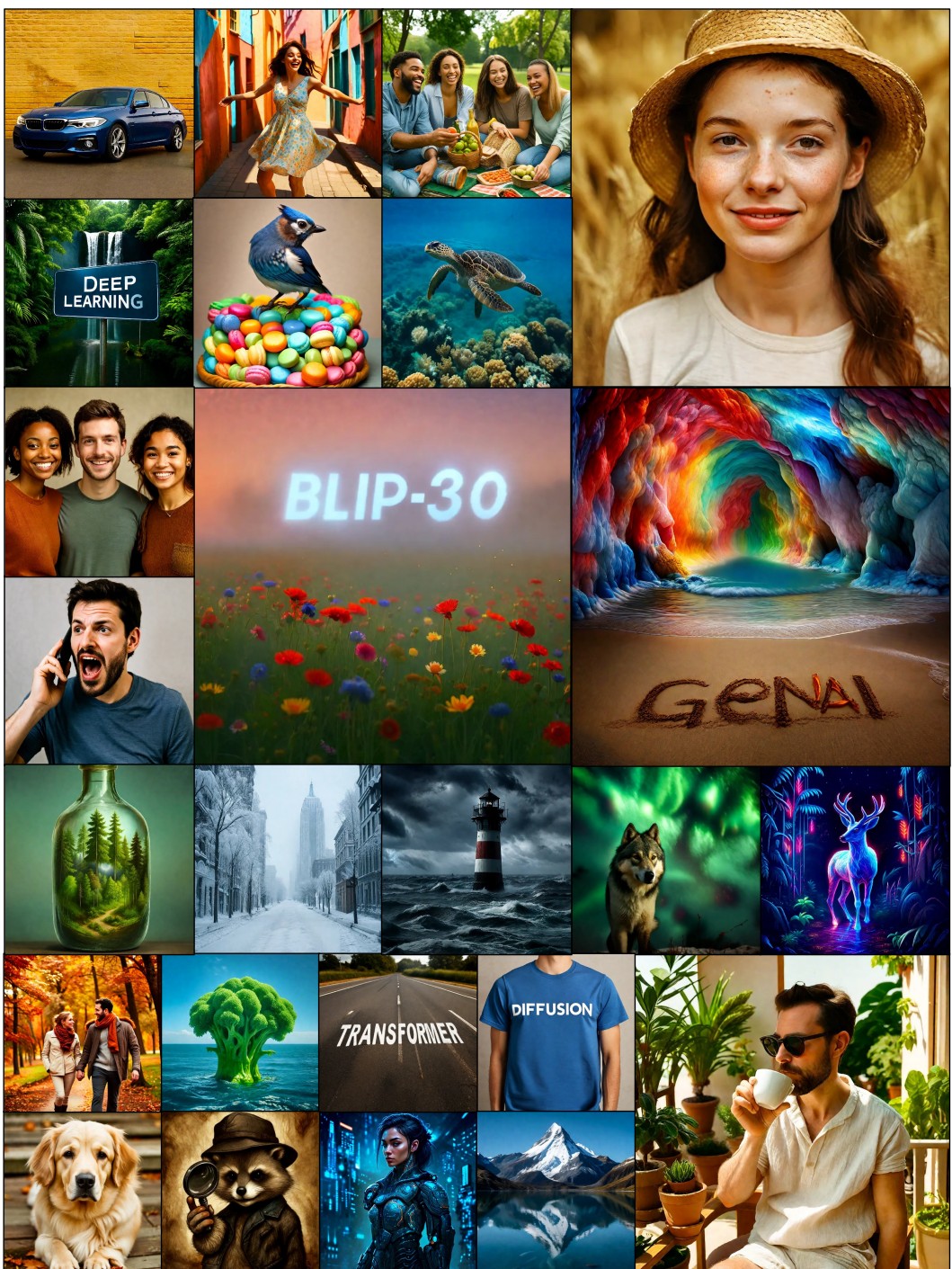

Figure 7: Visualization results of BLIP3-O 8B at 1024×1024 resolution.

## A PROMPT USED IN FIGURE 7

- A blue BMW parked in front of a yellow brick wall.

- A woman twirling in a sunlit alley lined with colorful walls, her summer dress catching the light mid-spin.

- A group of friends having a picnic.

- A lush tropical waterfall, 'Deep Learning' on a reflective metal road sign.

- A blue jay standing on a large basket of rainbow macarons.

- A sea turtle swimming above a coral reef.

- A young woman with freckles wearing a straw hat, standing in a golden wheat field.

- Three people.

- A man talking animatedly on the phone, his mouth moving rapidly.

- A wildflower meadow at sunrise, 'BLIP3o' projected onto a misty surface.

- A rainbow-colored ice cavern, 'Salesforce' drawn in the wet sand.

- A giant glass bottle filled with a miniature summer forest inside.

- Walk through of frozen streets of Manhattan, New York City—frozen trees and a frozen Empire State Building.

- A lighthouse standing alone in a stormy sea

- A lone wolf beneath shimmering northern lights.

- A glowing deer walking through a neon-lit futuristic jungle.

- A couple walking hand in hand through a vibrant autumn park, leaves gently falling around them.

- A curious vessel, shaped like a giant green broccoli, floating on a sparkling ocean under bright sunlight.

- 'Transformer' written on the road.

- 'Diffusion' on the blue T-shirt.

- A golden retriever lying peacefully on a wooden porch, with autumn leaves scattered around.

- A raccoon wearing a detective's hat, solving mysteries with a magnifying glass.

- A cyberpunk woman with glowing tattoos and a mechanical arm beneath a holographic sky.

- The reflection of a snowy mountain peak in a crystal-clear alpine lake, forming a perfect mirror image.

- A man sipping coffee on a sunny balcony filled with potted plants, wearing linen clothes and sunglasses, basking in the morning light.

## B  DIFFUSION TRANSFORMER ARCHITECTURE

We leverage the architecture of the Lumina-Next model (Zhuo et al., 2024) for our DiT. The Lumina-Next model is built on the improved Next-DiT architecture, a scalable and efficient diffusion transformer designed for text-to-image and general multimodal generation. It introduces 3D Rotary Position Embedding to encode spatial-temporal structure across time, height, and width without relying on learnable position tokens. Each transformer block employs sandwich normalization (RMSNorm before and after attention/MLP) and Grouped-Query Attention to enhance stability and reduce computation. Based on empirical results, this architecture achieves fast, high-quality generation.

## C  PRETRAINING DATA FOR IMAGE GENERATION

For 8B model, we combine around 25 million open-source data (CC12M (Changpinyo et al., 2021), SA-1B (Kirillov et al., 2023), and JourneyDB (Sun et al., 2023)) with an additional 30 million

proprietary images. All image captions are generated by Qwen2.5-VL-7B-Instruct, yielding detailed descriptions with an average length of 120 tokens. To improve generalization to varying prompt lengths, we also include around 10% (6 million) shorter captions with around 20 tokens from CC12M (Changpinyo et al., 2021). Each image–caption pair is formatted with the prompt: "Please generate an image based on the following caption: <caption>". For the fully open-source 4B model, we use 25 million publicly available images, from CC12M (Changpinyo et al., 2021), SA-1B (Kirillov et al., 2023), and JourneyDB (Sun et al., 2023), each paired with the same detailed captions. We also mix in around 10% (3 million) short captions sourced from CC12M (Changpinyo et al., 2021). **To support the research community, we release 25 million detailed captions and 3 million short captions.**

## D   HUMAN STUDY

In the human study, annotators compare image pairs side by side on two metrics:

- Visual Quality: the instruction is "All images were generated from the same text input using different methods. Please select the BEST image you prefer based on visual appeal, such as layout, clarity, object shapes, and overall cleanliness."

- Prompt Alignment: the instruction is "All images were generated from the same text input using different methods. Please select the image with the BEST image-text content alignment."

## E   THE USE OF LARGE LANGUAGE MODELS

During the preparation of this manuscript, large language models (LLMs) were used only as supportive tools for language refinement. Their role was limited to enhancing readability by improving sentence structure, correcting grammatical issues, and clarifying expression. They did not contribute to the development of research ideas, the design or execution of experiments, data analysis, or the creation of major content.

