# OpenReview forum: "BLIP3-o: A Family of Fully Open Unified Multimodal Models—Architecture, Training and Dataset"
_ICLR.cc/2026/Conference — ICLR 2026 Conference Withdrawn Submission_

### Official Review · Reviewer_u3fo · 2025-10-25

**Soundness:** 3
**Presentation:** 2
**Contribution:** 2
**Rating:** 2
**Confidence:** 3

**Summary:**

This paper introduces BLIP3-o, a family of unified multimodal models designed for both visual understanding and image generation. The proposed models represent images in the CLIP embedding space and generate images using a diffusion transformer trained with flow matching. The paper further investigates three key design dimensions: (1) CLIP vs. VAE latent representations, (2) flow matching vs. MSE regression, and (3) sequential vs. joint training for integrating understanding and generation. The experimental explorations demonstrate that the combination of CLIP embeddings, flow matching and sequential training results in a stable and high-performing unified framework.

**Strengths:**

The paper systematically evaluates important architectural and training choices for unified vision-language models, providing practical observations. The results show that sequential training prevents interference between understanding and generation tasks. The combination of CLIP embeddings and flow matching leads to a stable generation pipeline with strong text-image alignment. The release of datasets and models benefits research in unified multimodal modeling.

**Weaknesses:**

- The proposed pipeline closely resembles prior models and shows somewhat limited novelty in terms of technical architecture. The paper does not introduce a fundamentally new modeling paradigm, with the contributions lying primarily in empirical validation rather than conceptual innovation.

- While the paper highlights three design choices (CLIP embeddings, flow matching, and sequential training), these directions are familiar and intuitive in recent multimodal literature. The explorations in this paper primarily confirm existing intuitions rather than offer fundamentally new insights.

- The 8B model uses 30M well-curated private images, yet the experiments do not include thorough ablations on the training data. This makes it difficult to disentangle the performance improvements attributable to the applied approach from those arising from differences in data scale and quality.

- While human preference studies are central to the claims of improved visual quality, the work lacks transparency about human studies, including rater sourcing, annotation protocols, agreement metrics, and quality control procedures.

Overall, this work explores design choices for unified multimodal models and serves as a solid technical report from an engineering perspective. In particular, the open-sourcing of models and datasets is valuable and has a positive impact on the community. However, the paper lacks exploration of new frameworks or techniques, and most of the investigated conclusions about design choices are already known or intuitive. To meet the standards of top-tier conferences, the work is expected to present more technical innovations and deliver deeper scientific insights to advance the understanding of unified multimodal modeling.

**Questions:**

Please see weaknesses.

---

### Official Review · Reviewer_xR6L · 2025-10-30

**Soundness:** 3
**Presentation:** 3
**Contribution:** 2
**Rating:** 2
**Confidence:** 4

**Summary:**

This paper primarily discusses and experiments with unified multimodal large models from the perspectives of architecture, loss design, and training strategies. Under an autoregressive + diffusion framework, the paper finds that, in terms of generation architecture and loss formulation, the CLIP + FlowHead approach yields the best performance. Regarding training strategy, training the understanding component first followed by the generation component achieves optimal model performance. The paper ultimately demonstrates competitive results on both understanding and generation tasks.

**Strengths:**

1. The paper's presentation is excellent, clearly illustrating the method's architecture and the differences in the comparative experiments.
2. BLIP3o demonstrates competitive performance, showing strong results on both understanding and generation.
3. The paper proposes a method that effectively integrates open-source understanding and generation models to achieve a unified large model.

**Weaknesses:**

1. Paper does not provide a more in-depth discussion comparing its approach with EMU1 and EMU2. In my opinion, BLIP3O's architecture is very similar to that of EMU1/2, and its performance gains appear to stem primarily from the stronger base model.
2. GPT-4o not only excels at understanding and generation but also performs very well on image editing tasks. The paper mentions in Section 2 that the discussion and experiments are conducted under the assumption of a GPT-4o-based architecture. However, if focus solely on generation and understanding while ignoring editing, this might affect the conclusions of the paper. For example, for the choice of generation features, can using only high-level semantic features like those from CLIP still achieve competitive performance on editing tasks?
3. Insufficient experiments. Regarding the choice of generative features, the paper does not conduct experiments or comparisons on a CLIP + VAE + Flow Matching architecture. Such a design would preserve both high-level semantic information and low-level pixel information from the input. How would this affect the model’s understanding and generation capabilities? And could this also allow the model to better handle editing tasks?

**Questions:**

Please refer to the Weaknesses

---

### Official Review · Reviewer_324M · 2025-10-30

**Soundness:** 4
**Presentation:** 3
**Contribution:** 4
**Rating:** 6
**Confidence:** 3

**Summary:**

The paper systematically explores the design choices of training unified multimodal models, i.e. image representation, training objectives, and training strategies. There are some interesting experimental conclusions. (1)  CLIP features (with 64-dimensional compact semantic representation) outperform VAE latent features in both training efficiency and generation quality. (2)  Flow Matching loss (which models distribution transport via diffusion Transformers) addresses the lack of diversity in MSE loss. (3) sequential training (first pre-training the image understanding module, then freezing the backbone to train the diffusion generation module) outperforms joint training.

**Strengths:**

1. Systematic evaluation: The systematic validation of component combinations (e.g., why CLIP pairs better with Flow Matching than VAE) adds theoretical depth.​

2. High-quality experiments: The evaluation covers both understanding (9 benchmarks) and generation (automatic + human) tasks, with statistical significance tests (e.g., p-values in human evaluation) that strengthen result credibility.

3. Significance for practice: The open-source release promotes reproducibility and follow-up research.

**Weaknesses:**

1. The paper does not invent new components. The technical novelty is somewhat limited.

2. Lack of detailed ablations about w/ or w/o BLIP3o-60k training.

3. Insufficient comparison to recent recent related works about unified LMMs, such as Harmon [a] and Bagel [b].


[a] Wu S, Zhang W, Xu L, et al. Harmonizing visual representations for unified multimodal understanding and generation[C]. ICCV 2025.
[b] Deng C, Zhu D, Li K, et al. Emerging properties in unified multimodal pretraining[J]. arXiv preprint arXiv:2505.14683, 2025.

**Questions:**

Limited analysis of failure cases: For example, does BLIP3-O struggle with abstract concepts (e.g., "a painting of happiness") or low-data objects (e.g., rare animals)? Understanding limitations would strengthen the paper’s contribution.​

---

### Official Review · Reviewer_n1Rg · 2025-10-31

**Soundness:** 3
**Presentation:** 2
**Contribution:** 3
**Rating:** 6
**Confidence:** 4

**Summary:**

This paper systematically explores the architecture design space for unified multimodal models. Specifically, this work comprehensively studies the image representations and regression target. Based on the pilot studies, this paper adopts "CLIP + Flow Matching + two Diffusion Decoders" as the connected unified multimodal model. Extensive experimental results demonstrate that the proposed framework is able to reach state-of-the-art performance on image generation benchmarks.

**Strengths:**

1. The experiments are thorough and well-designed, covering a wide range of scenarios.

2. The model achieves competitive performance across standard image generation benchmarks.

3. The anticipated open-source release is valuable and likely to benefit the broader research and practitioner community.

**Weaknesses:**

1. A figure illustrating the overall model architecture, including the diffusion decoder that generates image pixels, is required. Otherwise, it is confusing when viewing Figure 1 (“The architecture of BLIP3-O”), which only presents the encoder without showing the decoder architecture for image generation.

2. This work overlooks the entanglement between the proposed architecture, the GPT-4o synthetic data, and the pre-trained Lumina-Next model. Given that synthetic data can significantly boost performance on benchmarks such as GenEval, it is unclear whether the improvements stem from the architecture itself, the synthetic data, or the strong pre-trained backbone. Moreover, without controlled experiments on pre-trained Lumina-Next, it remains possible that Lumina-Next trained with similar data could achieve comparable results.

3. This raises the question of whether such unification is necessary. To address this concern, the paper should demonstrate capabilities that bespoke generative models cannot achieve. Therefore, the current benchmarks are not fully convincing in motivating the unification.

4. What about the potential of this stage-wise training paradigm versus the end-to-end one, and what are the weaknesses?

**Questions:**

See weaknesses. I would adjust my rating according to the authors' response.

---

### Note · Authors · 2025-11-19

I have read and agree with the venue's withdrawal policy on behalf of myself and my co-authors.